# Molecular Features and Actionable Gene Targets of Testicular Germ Cell Tumors in a Real-World Setting

**DOI:** 10.3390/ijms26188963

**Published:** 2025-09-15

**Authors:** Rafael Morales-Grimany, Krinio Giannikou, Cesar Delgado, Kshitij Pandit, Fady Baky, Armon Amini, Kit Yuen, Thomas Gerald, Rohit Badia, Jacob Taylor, Luke Wang, Juan Javier-Desloges, Vitaly Margulis, Solomon Woldu, Amirali Salmasi, Fred Millard, Rana R. Mckay, Aditya Bagrodia

**Affiliations:** 1Department of Medicine, School of Medicine, Universidad Central del Caribe, Bayamon, PR 00956, USA; 121rmorales@uccaribe.edu; 2Department of Urology, Moores Cancer Center, School of Medicine, University of California San Diego, La Jolla, CA 92093, USA; kgiannikou@health.ucsd.edu (K.G.); cedelgado@health.ucsd.edu (C.D.); kpandit@health.ucsd.edu (K.P.); k3yuen@health.ucsd.edu (K.Y.); llw005@health.ucsd.edu (L.W.); jjavierdesloges@health.ucsd.edu (J.J.-D.); asalmasi@health.ucsd.edu (A.S.); 3Department of Urology, School of Medicine, University of Texas Southwestern, Dallas, TX 75390, USA; fady.baky@phhs.org (F.B.); armon.amini@utsouthwestern.edu (A.A.); thomas.s.gerald.mil@health.mil (T.G.); rohit.badia@utsouthwestern.edu (R.B.); jacob.taylor.mail@gmail.com (J.T.); vitaly.margulis@utsouthwestern.edu (V.M.); solomon.woldu@utsouthwestern.edu (S.W.); 4Department of Medicine, School of Medicine, University of California San Diego, La Jolla, CA 92093, USA; fmillard@health.ucsd.edu (F.M.); rmckay@health.ucsd.edu (R.R.M.)

**Keywords:** chemoresistance, seminoma, teratoma, embryonal carcinoma, mutations, targeted therapy

## Abstract

Molecular profiling of testicular germ cell tumors (TGCTs) provides critical insights into personalized treatment approaches, particularly for patients with recurrent or treatment-resistant disease. In this study, we retrospectively analyzed clinicopathological and targeted genomic sequencing data from 27 TGCT patients, including 7 seminomas, 19 non-seminomas, and 1 prepubertal type teratoma, across stage I (48%), stage II (41%), and stage III (11%). Tumor samples were obtained from 27 orchiectomies, with additional pathological specimens collected from 16 of these patients during retroperitoneal lymph node dissections (RPLNDs); these included 8 chemotherapy-naïve and 8 post-chemotherapy cases. The median tumor mutational burden (TMB) was 0.5 mutations/Mb, consistent with the low mutation rate typically observed in TGCTs. Somatic mutations and copy number gain alterations were detected in 56% (15/27) of patients, primarily in *KRAS* (25.9%), *KIT* (11.1%), and *PIK3CB* (7.4%). PD-L1 positive immunoreactivity by immunohistochemistry was observed in 75% of tumors (60% in stage I, 100% in stage III) analyzed (*n* = 8), suggesting potential immune checkpoint inhibitor applicability in advanced disease. Microsatellite instability (MSI) status was identified in 23 tumors; all were classified as MSI-low, supporting the rarity of MSI-driven tumorigenesis in TGCTs. Actionable gene alterations linked to FDA-approved therapies, interventional therapies, and clinical trials in TGCTs and other cancers (lung, skin, colon, liver, stomach, and breast) were present in 59.3% (16/27) of patients, indicating potential therapeutic repurposing. Additionally, germline variants of uncertain clinical significance in known cancer actionable genes, including *MSH2*, *MSH6*, *RB1*, and *BRCA2*, were found in 9 patients, warranting further investigation regarding their clinical relevance and susceptibility risk. Our findings highlight that a substantial proportion of TGCT patients harbor potentially actionable molecular alterations across all disease stages.

## 1. Introduction

Testicular germ cell tumors (TGCTs) are the most common malignancy in young men aged 15–34, with a prevalence of 309,202 patients in the United States of America [1]. Despite this, the genomic landscape of TGCTs has been under-described compared to other solid organ tumors [2]. The 5-year survival rate for localized, regional, and distant stages of testicular cancer is 90–95%, with approximately 10% of these patients developing refractory disease, which has been associated with significantly higher mortality rates [3]. This highlights the need for a more comprehensive characterization of the genetic drivers of TGCTs to improve the current therapeutic strategies and potentially uncover new targeted therapies in these high-risk patients [4].

The high survival statistics can be attributed to cisplatin-based chemotherapy, which has been the cornerstone of testicular cancer treatment, achieving remarkable success in curing the majority of patients [5]. However, resistance to cisplatin remains a significant challenge. Some of these resistance mechanisms have already been observed in other tumors, and successful therapies have already been implemented [5,6,7,8,9]. Copper chelators in bladder cancers have shown downregulation of ATP7B transporters, restoring cisplatin levels [6]. In ovarian cancers, anti-glutathione strategies with buthionine sulfoximine have been shown to prevent cisplatin detoxification [7]. Moreover, ATK/CHK1 inhibitors in ovarian cancer decrease DNA repair activity, preventing the repair of cisplatin’s DNA damage [9]. PD-1/PD-L1 inhibitors have been used in lung cancers to increase T-cell activation and induce cell death [8]. These molecular adaptations, which are also seen in TGCT cohorts, have resulted in these patients’ treatment failures and deaths [10]. Combining this adversity with the long-term toxicities associated with cisplatin and other chemotherapeutic agents—including nephrotoxicity, ototoxicity, neurotoxicity, and cardiovascular complications—have significantly impacted survivors’ quality of life [11]. As a result, current research has focused on identifying chemotherapies with an improved safety profile and integrating the understandings from other platinum-resistant cancers to pave the way for novel anti-neoplastic therapies for TGCTs.

Current advances in genomic technologies, such as DNA sequencing and transcriptomic profiling, have revolutionized cancer research, and the integration of molecular profiling into clinical settings has greatly expanded our understanding of the genomic landscape in many cancers, including TGCTs [12]. These developments have facilitated the emergence of targeted therapies and personalized treatment approaches. However, despite these advances, testicular cancer remains relatively understudied in terms of comprehensive genomic analyses, particularly within diverse and large real-world patient cohorts [13].

Management of TGCTs depends on histologic subtype and clinical stage, with stage I patients having multiple treatment options [14]. Seminomas are often managed with surveillance or adjuvant radiotherapy, while non-seminomatous germ cell tumors (9 s) may be treated with surveillance, adjuvant chemotherapy, or retroperitoneal lymph node dissection (RPLND) [14]. Although risk-adapted strategies vary between institutions, the National Comprehensive Cancer Network (NCCN) guidelines provide a general framework [14]. In this study, we include patients managed at a tertiary cancer center, where treatment decisions are individualized based on tumor markers, imaging, histology, and shared decision-making within a multidisciplinary team.

The objective of this study was to characterize the landscape of somatic alterations in TGCTs using targeted gene sequencing and to determine the potential clinical relevance of these genomic alterations. Specifically, we aimed to identify actionable mutations and evaluate their distribution across histological subtypes, with a focus on understanding the feasibility and implications of precision oncology approaches in this rare tumor type.

## 2. Results

The cohort of 27 TGCT patients analyzed had a median age of 32 years (age range 23–61 years; IQR: 11). Overall, 13 (48%) had stage I GCT, 11 (41%) stage II, and 3 (11%) stage III disease at diagnosis, with a median follow-up of three years. Seven tumors were seminomas, 19 were non-seminoma, and one pre-pubertal type teratoma (Figure 1A, Table 1, Appendix A). All 27 patients underwent orchiectomy, and 16 patients also underwent RPLND, with 10 having RPLND pathology (Table 1). Of the RPLND patients, eight were chemotherapy-naïve, and eight were post-chemotherapy residuals (Figure 1B). The chemo-naïve RPLND subset included two seminomas, five NSGCTs, and one pre-pubertal type teratoma. In contrast, the post-chemotherapy RPLND subset revealed eight NSGCT (Table 1, Figure 1B). Pre-orchiectomy serum levels of AFP, beta-hCG, and LDH were measured in the majority of cases, demonstrating broad variability across patients (Appendix A). AFP levels ranged 0.6–46,887 ng/mL, beta-hCG levels were 1–14,240 ng/mL, and LDH levels were 112–2090 ng/mL. Adjuvant postoperative chemotherapy was administered to one patient (TGCT-13, mixed NSGCT) after RPLND revealed high volume viable TGCT.

TMB had a median value of 0.5 mutations/Mb and a mean of 0.75 mutations/Mb across all TGCT samples. Reflecting the low somatic mutation rate that characterizes TGCTs in general (Figure 1C). No statistical difference was identified in the TMB between seminomas and non-seminomas. Somatic actionable variants were identified in 15/27 (56%) of patients with variant allele frequency in a range of 2.12–59.7% in 20 mutated genes. The most recurrent molecular alterations included *KRAS*, *KIT*, and *PIK3CB* alterations in 25.9% (7/27), 11.1% (3/27), and 7.4% (2/27) of all tumors, respectively (Figure 1D).

Within our cohort, there were four somatic *KRAS* mutations, all of which were missense gain of function variants, and four copy number gain alterations in *KRAS*. All four *KRAS* missense mutations, including p.G12R (TGCT-3), p.G12C (TGCT-8), p.G12V (TGCT-10), and p.G12D (TGCT-24) were located in exon 2. Out of these, only the *KRAS* p.G12C mutation has specific FDA-approved therapy with sotorasib or adagrasib. The *KRAS* p.G12R, p.G12V, and p.G12D alterations have guideline-endorsed predictive responses to FDA-approved drugs as reported by the National Comprehensive Cancer Network with MEK inhibitors (trametinib and cobimetinib). Notably, all four *KRAS* missense mutations have reported resistance to EGFR inhibitors (cetuximab, panitumumab) and HER2 inhibitors (tucatinib + trastuzumab) for various cancers.

Four *KRAS* alterations were present in three mixed germ cell tumors (TGCT-10, TGCT-24, TGCT-25). One mixed germ cell tumor case exhibited both a *KRAS* missense variant p.G12D and a copy number gain alteration for *KRAS* (TGCT-24). A total of 50.0% (1/2) pure teratomas had a *KRAS* missense mutation p.G12C, and the only pure embryonal carcinoma had a KRAS copy number gain (TGCT-18). Also, 28.9% (2/7) of seminomas had *KRAS* alterations. One seminoma had a *KRAS* copy number variant, which co-existed with a *KIT* p.N822K missense mutation (TGCT-7). The singleton missense gain of function variant (p.E1051K) identified in *PIK3CB* was observed in two seminoma samples (TGCT-2 and TGCT-3), where it co-existed with either *KIT* (p.N822K) or *KRAS* (p.G12R) mutations.

Additionally, all *KIT* alterations were seen in seminomas only. A total of 42.9% (3/7) of seminomas had a *KIT* mutation, with missense gain-of-function variants detected in three tumors (TGCT-2, TGCT-6, and TGCT-7). TGCT-2 and TGCT-7 had the same amino acid alterations in *KIT* (p.N822K). Four *KIT* alterations were identified, with one seminoma (TGCT-7) having a co-existing missense mutation and a copy number gain. Out of 15 known cancer-actionable genes with curated drug resistance data reported in COSMIC database, *KIT* was the only mutant gene in this cohort with two different missense variants known to be chemoresistant (Table 2). The missense mutation *KIT* p.N822K (GRCh37/hg19) and p.D816V have been associated with resistance to tyrosine kinase inhibitors Imatinib and Sunitinib. Nonetheless, there are guideline predictive responses to these mutations with FDA-approved therapies from expert committees like the NCCN, with regorafenib and ripretinib. Regorafenib has a broader spectrum due to its increased flexibility to many amino acid alterations in the KIT protein conformation, and ripretinib is a switch pocket inhibitor, a critical allosteric site for regulating enzyme activity. Notably, over half of our study’s patients had molecularly targetable alterations, which included FDA-approved therapies for other cancers (lung, stomach, skin, colon), interventional therapies, or clinical trials.

In addition, MSI status was assessed in all 27 patients. The MSI results for four patients were not reported due to <30% tumor content (Appendix A). Of the remaining twenty-three patients, all were classified as MSI-stable. PD-L1 immunohistochemistry (IHC) staining was performed on eight tumors, including two seminomas and six non-seminomas, which demonstrated overall positivity of 75% of the tumors measured, with a 60% (3/5) positivity at stage I and 100% (3/3) positivity at stage III (Appendix A).

Within our cohort, we identified ten germline variants of unknown significance (VUS) in nine patients. Most of them were missense, affecting well-established cancer-associated genes such as DNA mismatch repair genes (*MSH2*, *MSH6*) and tumor suppressor genes, e.g., *BRCA2* and *RB1* (Table 3). Additionally, 15 somatic VUS were detected in 13 patients, including key tumor suppressor genes such as *KMT2C*, *MTOR*, *NF1*, and *NF2*, with a median variant allele frequency of 12.8% (range: 5.4% to 28.27%) (Table 3). Most of these variants are listed in COSMIC, Varsome, and/or Clinvar databases with limited clinically significant data regarding pathogenicity.

## 3. Discussion

In this study, we performed targeted gene sequencing and identified key molecular features of TGCTs, including frequent *KRAS*, *KIT*, and *PIK3CB* mutations consistent with previous literature [19]. Our genomic analysis also confirmed previous reports of low TMB, recurrent *KIT* and *KRAS* mutations, and lack of MSI [20]. To further characterize this study, we compared our results with a larger cohort study of 137 TGCT patients [21]. Both studies identified *KRAS* and *KIT* as the most frequently altered genes in TGCTs [21]. Our study observed a higher *KRAS* alteration rate (25.9% vs. 14.0%), whereas the other study reported a slightly higher *KIT* mutation rate (18.0% vs. 11.1%) [21]. The cohort sizes differed by a total of 110 patients (27 vs. 137). Despite the larger cohort, all but one of the *KRAS* somatic mutations in that study were exclusive to seminomas, whereas we identified six *KRAS* alterations in non-seminomatous TGCTs. Both studies found *KIT* mutations exclusively in seminomas. Mutations in the *PIK3C* family were also observed in both studies, restricted to seminomas. Shen et al., 2018 identified alterations in *PIK3CA* and *PIK3CD* [21], while our study detected mutations only in *PIK3CB*. Additionally, they found a small but significant association between *NRAS* mutations and seminomas, a finding not replicated in our cohort. Both studies had an identical median TMB of 0.5 mutations per Mb. Most notably, our study identified several *KRAS* mutations in mixed GCTs and NSGCTs, a key distinction from the study by Shen et al. 2018 [21]. The genetic alterations identified in these studies contribute to the genomic landscape for risk stratification and the development of molecularly informed treatment strategies for TGCT patients. Furthermore, the high PD-L1 expression observed in advanced-stage tumors suggests potential applications for immunotherapy [22].

The genomic landscape of TGCTs is characterized by a relatively low TMB compared to other solid tumors, with frequent chromosomal abnormalities such as gains in chromosome 12p and loss of heterozygosity in specific regions, e.g., chr1p, 11q, 13q, 18q [23,24]. TGCTs often present somatic mutations in *KIT* and *KRAS*, as well as gene components of the PI3K/AKT/mTOR pathway [23,25]. Epigenetic alterations, including global DNA hypomethylation and specific promoter hypermethylation, are also prominent features, particularly in cisplatin-resistant tumors [2]. Despite advances in identifying these alterations, integrating genomic findings into clinical practice remains limited. Clinical trials for renowned KIT inhibitors (imatinib and sunitinib) in TGCTs have demonstrated limited activity due to potential disease reliance on genetic drivers and not singleton mutations [26]. *KRAS* mutations have also been reported to express resistance specifically to EGFR inhibitors (cetuximab, panitumumab) and HER2 inhibitors (tucatinib + trastuzumab) [27,28]. Both types of treatments converge on suppressing the RAS-RAF-MAPK pathway by targeting upstream transmembrane receptors [27,28]. However, when *KRAS* is mutated, the protein enters a perpetually activated state due to a permanently locked formation with GTP. This mutation results in constitutive activation of downstream signaling, making it partially or mostly independent from HER2 and EGFR signaling [27,28]. This underscores the need for further studies to identify new actionable/druggable targets and develop precise therapies for TGCTs. Definitions of cohort selection, clinical variables, and outcomes were clarified to ensure transparency and reproducibility.

Another critical reason to identify more specific targets and develop equally effective novel treatments is to address the significant side effect profile posed by bleomycin, etoposide, and platinum. Despite success, chemotherapy is reported to cause significant side effects in more than half of patients [29]. Adverse effects range from pulmonary toxicity two to four times more severe than smoking, increased risk of secondary malignancies, kidney injury, cardiovascular damage, and sexual dysfunction. The severity of these effects directly impacts the quality of life of TGCT patients. Interventional lifestyle strategies, including exercise, smoking cessation, a healthy diet, and managing underlying conditions, have been shown to decrease quality of life impairment [30]. Currently, approaches to de-escalating therapies are being pursued. Trials are underway involving RPLND in stage II seminoma patients instead of chemotherapy and decreasing chemotherapy dosing to mitigate the severity of side effects while preserving efficacy [31,32,33]. Although significant, these trials present limitations in addressing various types of TGCT and must be conducted in centers equipped for multimodal therapy. This highlights the relevance of exploring targeted therapies and immunotherapies as potential strategies to reduce treatment related adverse effects and mortality, especially in poor prognosis cases. Despite this, comprehensive chemotherapy remains part of the gold standard in the management of testicular neoplasms, with a continued focus on maintaining high remission rates while emphasizing the importance of quality of life.

Among the variants of uncertain significance (VUS) identified in this cohort, several were found in genes with known implications for cancer risk and therapeutic response, including *BRCA2*, *MSH2*, *MSH6*, and *RB1*. While these variants are not currently classified as pathogenic, their occurrence in TGCT patients raises important questions regarding possible roles in disease predisposition or treatment resistance. For example, *BRCA2* alterations have been associated with platinum sensitivity in prostate and ovarian cancers, and mismatch repair gene variants (*MSH2*, *MSH6*) are relevant to immunotherapy responses in other tumor types. In our cohort, all patients were microsatellite-stable, and no consistent associations were observed between these VUS and clinical outcomes, tumor stage, or family history, likely due to the small cohort size and retrospective design. We have annotated these variants in Table 3, which is a supplementary ACMG-style classification table, and emphasize the need for future studies with larger patient cohorts and longer clinical follow-up. Future functional validation using in vitro GCT models, DNA damage response assays, and pedigree analysis will be critical to clarify the biological and clinical relevance of these findings.

This study has several limitations. First, we acknowledge the relatively small sample size (*n* = 27) analyzed from a histologically diverse patient cohort, which limited genotype–phenotype correlations and subgroup analyses. Second, details such as history of cryptorchidism, hypospadias, infertility, marijuana use, pesticide exposure, post-orchiectomy tumor values, maternal smoking, medications during pregnancy, and family history of testicular cancer data were not uniformly available due to the retrospective nature of the data collection. Third, while the use of the Tempus xT panel provides valuable genomic insights, it is restricted to a limited paired sample set of 648 genes, potentially overlooking other relevant genomic alterations, e.g., copy number alterations or non-coding regions of the genome. Fourth, we recognize that although FDA-approved therapies for *KIT* and *KRAS* mutations are described, no functional validation involving gene expression, signaling pathway, or in vitro drug-sensitivity analyses was performed, and we highlight this as an important direction for future research. Functional validation with these tests is an important direction for future research. Additionally, studies in larger, more diverse patient cohorts, single-cell and other multi-omics approaches, and functional assays are warranted to confirm and expand these findings.

In conclusion, this study highlights the molecular heterogeneity of TGCTs across disease clinical states and emphasizes the importance of molecular profiling to identify actionable targets and inform suitable therapeutic strategies, particularly for patients with recurrent, refractory, or treatment-resistant TGCTs.

## 4. Materials and Methods

### 4.1. Patient Cohort and Methods

Following the Institutional Review Board (IRB#190443) approval of our study, we collated clinicopathological and molecular/genomic data from 27 patients with TGCTs seen at the Department of Urology, University of Southwestern, Texas, between 2018 and 2021. Diagnosis was made based on clinicopathologic/morphologic and immunohistochemical characteristics evaluated by pathologists with expertise in TGCTs according to the 2022 WHO Classification of Germ Cell Tumors [34]. All 27 TGCTs had a tumor-cell content estimated to be ≥20% (range 20–90%, median = 50%) and necrosis < 10%. Genomic DNA from formalin-fixed paraffin-embedded (FFPE) tumor tissue and matched normal (saliva or blood) tissue was analyzed for all cases. All tumor specimens were analyzed using Tempus xT panel version 4 (Tempus Labs, Chicago, IL, USA), which covers 648 cancer actionable target genes spanning approximately 3.6 Mb of genomic DNA, translocations in 22 genes, along with two promoter regions (*PMS2* and *TERT*), and 239 sites to determine microsatellite instability (MSI) status [35]. The study was conducted in compliance with the Helsinki Declaration.

In accordance with Strengthening the Reporting of Observational Studies in Epidemiology (STROBE) guidelines and the Preferred Reporting of Case Series in Surgery (PROCESS 2023), this was a single-center, retrospective analysis of consecutively sequenced cases seen at UTSW between 2018 and 2021 [36,37]. Patients included in this cohort were managed at UT Southwestern, Texas. Initial treatment for stage I seminoma typically involves surveillance or single-agent carboplatin. For stage I NSGCT, treatment options included surveillance, adjuvant bleomycin-etoposide-cisplatin (BEP), or RPLND. Our institutional approach follows NCCN guidelines but also considers tumor marker kinetics, histopathologic risk factors (e.g., lympho-vascular invasion), and patient preferences when selecting treatment modality. RPLND was performed selectively for patients with high-risk features or residual masses post-chemotherapy. The biological classification of TGCTs into germ cell neoplasia in situ (GCNIS)-associated and non-GCNIS-associated tumors has important implications for understanding pathogenesis and clinical behavior. In our cohort, the well-differentiated neuroendocrine tumor (TGCT-20), consistent with a prepubertal-type teratoma, was considered non-GCNIS-associated based on histology, absence of germ cell neoplasia in situ, and clinical presentation. This tumor was analyzed separately and excluded from aggregate molecular summaries for postpubertal GCNIS-associated tumors. This distinction aligns with current WHO classifications and reflects divergent developmental origins and genetic profiles.

The primary outcome of this study was the identification of potentially actionable genomic alterations in TGCTs, defined by the presence of somatic mutations with therapeutic or diagnostic relevance according to OncoKB [38,39] and NCCN guidelines. Variables evaluated included patient demographics (age, race), clinical stage, histology, tumor mutational burden (TMB), PD-L1 expression, and the presence or absence of germline or somatic variants in a predefined gene panel. This data was used to explore genotype–phenotype correlations and assess mutation profiles across tumor subtypes. Statistical analyses were performed using R version 4.2.2 (Posit, PBC, Boston, MA, USA). Quantitative variables were defined and analyzed accordingly, and comparative analyses between groups were conducted using the Wilcoxon rank-sum test.

### 4.2. Gene Variant Call Analysis

Data from the National Comprehensive Cancer Network Guidelines [14] and the FDA-recognized section of OncoKB [38,39] were implemented to assess the pathogenicity of genetic variants and their relevance to matched therapies or biological significance. In silico tools were utilized for variant classification, including SIFT [40] and PROVEAN [41], and predicted splice variants were assessed using ADA (Adaptive Boosting) [42] and RF (Random Forest) [43]. Tumor Mutational Burden quantified the total number of somatic single-nucleotide variants (SNVs) and small insertions/deletions (indels) present in a tumor, regardless of their pathogenicity, including benign alterations. TMB was estimated as the number of protein-altering mutations per million coding base pairs.

## Figures and Tables

**Figure 1 ijms-26-08963-f001:**
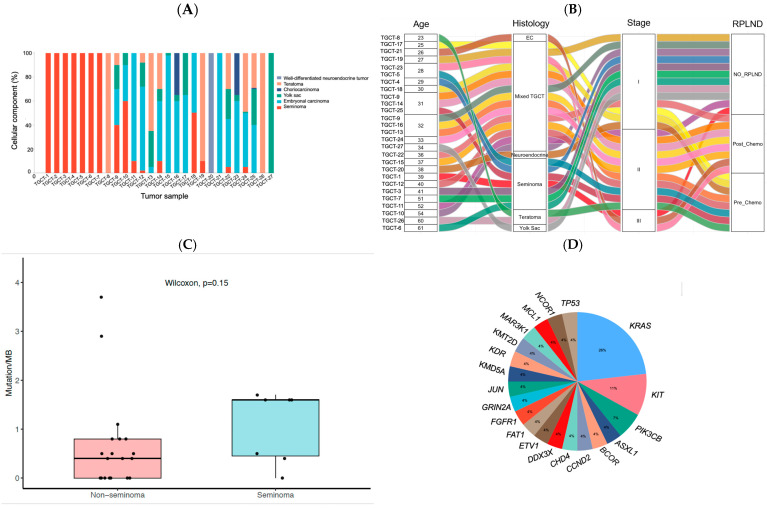
Molecular and clinicopathological overview of testicular germ cell tumors (TGCTs) in a real-world cohort (*n* = 27). (**A**) Histological subtype distribution across the cohort. Bar plot showing the number of patients diagnosed with TGCTs. Histology was based on pathology review of orchiectomy and/or RPLND specimens. TGCT-20 is a pre-pubertal type teratoma (well-differentiated neuroendocrine tumor) (**B**) Alluvial plot integrating clinicopathological features. This visualization depicts the relationship among multiple variables for each patient, including sample ID, patient age, histology (seminoma vs. non-seminoma), clinical stage (I, II, III), and whether retroperitoneal lymph node dissection (RPLND) was performed. This integrative view highlights histologic and clinical heterogeneity across the cohort. (**C**) Tumor Mutational Burden (TMB) stratified by histology. Boxplot showing the distribution of TMB (mutations per megabase) for seminomas versus non-seminomas. Median TMB for the entire cohort was 0.5 mutations/Mb, reflecting the overall low mutation rate in TGCTs. The Wilcoxon rank-sum test showed no significant difference between the two histologic subgroups (*p* = 0.15). (**D**) Prevalence of actionable somatic alterations. Bar chart illustrating the number of tumors harboring recurrent cancer-related mutations identified through targeted sequencing. The most frequently altered genes included *KRAS*, *KIT*, and *PIK3CB*. Only genes with at least two altered cases are shown. These alterations include both single-nucleotide variants and copy number changes with potential clinical significance based on NCCN and OncoKB databases. EC: Embryonal carcinoma. TGCT-1: seminoma with large amount of syncytiotrophoblast.

**Table 1 ijms-26-08963-t001:** Clinicopathological data in the Testicular Cancer Cohort.

Clinicopathological Variables		Number of Patients (%)
Clinical disease stage	I	13 (48)
	II	11 (41)
	III	3 (11)
Type of TGCT	Seminoma	7 (26)
	Non-Seminoma	19 (70)
	Pre-pubertal teratoma	1 (4)
Surgeries Performed	Orchiectomy	27 (100)
	RPLND	16 (59)
Chemotherapy status prior RPLND	Chemotherapy-naïve	8 (50)
	Post-chemotherapy	8 (50)

RPLND: Retroperitoneal Lymph Node Dissection.

**Table 2 ijms-26-08963-t002:** Summary of pathogenic somatic variants identified in the testicular tumor cohort according to Tempus.

Patient ID	Gene	AA Change	Alteration Effect	Function	Variant Allele Frequency	Mutations/MB
TGCT-1	*BCOR*	p.S177fs	Frameshift	LOF	10.50%	1.6
TGCT-2	*KIT*	p.N822K ^#^	Missense	GOF	8.10%	1.6
	*PIK3CB*	p.E1051K	Missense	GOF	8.60%	
TGCT-3	*KRAS*	p.G12R	Missense	GOF	4.00%	1.6
	*PIK3CB*	p.E1051K	Missense	GOF	8.60%	
TGCT-6	*KIT*	p.D816V ^#^	Missense	GOF	12.70%	1.7
TGCT-7	*KDR*	--	Copy number gain	--	--	0.4
	*KIT*	--	Copy number gain	--	--	
		p.N822K ^#^	Missense	GOF	39.36%	
	*KRAS*	--	Copy number gain	--	--	
TGCT-8	*KRAS*	p.G12C	Missense	GOF	59.70%	3.7
TGCT-10	*KRAS*	p.G12V	Missense	GOF	29.40%	0.8
TGCT-12	*GRIN2A*	--	Copy number loss	--	--	0.0
	*MCL1*	--	Copy number gain	--	--	
TGCT-14	*TP53*	p.L252_I254del	Inframe deletion	LOF	16.60%	0.5
TGCT-18	*ETV1*	--	Copy number gain	--	--	0.8
	*FGFR1*	--	Copy number gain	--	--	
	*JUN*	--	Copy number gain	--	--	
	*KRAS*	--	Copy number gain	--	--	
	*NCOR1*	p.Stgc43	Stop codon	LOF	2.12%	
TGCT-21	*DDX3X*	p.766-1G>T	Splice region	LOF	4.90%	0.0
	*KMT2D*	p.G5189 *	Stop codon	LOF	2.90%	
	*MAP3K1*	p.G331 *	Stop codon	LOF	3.70%	
TGCT-24	*CCND2*	--	Copy number gain	--	--	0.8
	*CHD4*	--	Copy number gain	--	--	
	*KDM5A*	--	Copy number gain	--	--	
	*KRAS*	--	Copy number gain	--	--	
		p.G12D	Missense	GOF	59.30%	
TGCT-25	*KRAS*	--	Copy number gain	--	--	0.4
TGCT-26	*ASXL1*	p.E565fs	Frameshift	LOF	16.56%	0.4
TGCT-27	*FAT1*	p.L1889fs	Frameshift	LOF	24.65%	2.9

LOF: loss of function, GOF: gain of function, AA: amino acid, ^#^ Chemoresistant variant (COSMIC database), *: Stop codon.

**Table 3 ijms-26-08963-t003:** Summary of variants of unknown significance (VUS) identified in 32 genes in the testicular tumor cohort.

Patient ID	Type of Variant	Gene	AA Alteration	Variant Effect	Variant Allele Frequency	ACMG Classification
TGCT-1	Somatic	*PRKN*	p.L102F	Missense	9.7%	VUS
	Somatic	*RANBP2*	p.P162A	Missense	15.2%	VUS
TGCT-2	Somatic	*C8orf34*	p.L523del	Inframe deletion	6.9%	VUS
	Germline	*MSH6*	p.T999P	Missense	N.A.	Likely Pathogenic
	Germline	*PMS2*	p.Q160H	Missense	N.A.	VUS
TGCT-3	Somatic	*KEL*	p.H581P	Missense	6.2%	VUS
	Germline	*PALB2*	p.V836I	Missense	N.A.	Likely Pathogenic
TGCT-4	Somatic	*FAM46C*	p.N18S	Missense	5.4%	N.A
TGCT-6	Somatic	*KMT2C*	p.C3460G	Missense	9.1%	Likely Pathogenic
	Somatic	*MAF*	p.H187del	Inframe deletion	6.0%	VUS
	Somatic	*MTOR*	p.K1981E	Missense	12.8%	Likely Pathogenic
TGCT-7	Germline	*MUTYH*	p.L420M	Missense	N.A.	Pathogenic
TGCT-10	Germline	*RET*	p.R180Q	Missense	N.A.	VUS
TGCT-11	Germline	*APOB*	p.Y3295H	Missense	N.A.	VUS
	Germline	*MSH6*	p.P66L	Missense	N.A.	Likely Pathogenic
	Somatic	*NF1*	p.L762M	Missense	13.9%	Likely Pathogenic
TGCT-13	Somatic	*CUX1*	p.R1374L	Missense	10.4%	N.A
	Somatic	*KMT2D*	p.R1918P	Missense	12.1%	N.A
TGCT-15	Somatic	*ERBB2*	p.E992K	Missense	25.9%	Pathogenic
TGCT-16	Germline	*APOB*	p.M1150K	Missense	N.A.	N.A
	Germline	*BRCA2*	p.F2058I	Missense	N.A.	N.A
	Germline	*RB1*	p.E137D	Missense	N.A.	N.A
TGCT-17	Germline	*BRCA2*	p.Q2858K	Missense	N.A.	N.A
TGCT-18	Somatic	*NF2*	p.T581I	Missense	25.5%	Likely Pathogenic
TGCT-20	Somatic	*IL7R*	p.G424V	Missense	7.4%	N.A
TGCT-24	Germline	*APOB*	p.G753E	Missense	N.A.	N.A
	Germline	*APOB*	p.Y129C	Splice site	N.A.	N.A
	Somatic	*JAK3*	p.R210W	Missense	27.8%	Likely Pathogenic
	Germline	*MSH2*	p.L119S	Missense	N.A.	N.A
	Germline	*MSH6*	p.E277D	Missense	N.A.	N.A
TGCT-25	Somatic	*GRM3*	p.C127Y	Missense	15.3%	VUS
TGCT-26	Germline	*RET*	p.E623K	Missense	N.A.	N.A
TGCT-27	Somatic	*HSP90AA1*	p.V266I	Missense	17.4%	N.A
	Somatic	*RUNX1*	p.S167I	Missense	20.5%	N.A
	Somatic	*WRN*	p.Q253 *	Stop codon	28.3%	Pathogenic

AA: amino acid, N.A: not applicable. *: Stop codon. Germline classification was assigned based on the ACMG guidelines [15] and somatic classification also incorporated current evidence from variant type (e.g., truncating or missense), known cancer hotspots, and annotations from curated databases including COSMIC [16], OncoKB, Varsome [17], and ClinVar [18], when available.

## Data Availability

The additional data supporting the manuscript are available from the corresponding author upon request.

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
