# Peer review of "Molecular Features and Actionable Gene Targets of Testicular Germ Cell Tumors in a Real-World Setting"

_ijms, 2025, doi:10.3390/ijms26188963_

Round 1
Reviewer 1 Report
Comments and Suggestions for Authors
1. Patients with testicular tumors may be managed with different treatment strategies even at the same stage. For example, in non‑seminomatous germ cell tumors, even at stage I there are options of surveillance, chemotherapy, and RPLND. Although these are fundamental matters, considering that the readers of this journal are not necessarily specialists in testicular tumors, it might be helpful, in the Introduction section or the Materials and Methods section, to state concisely the general treatment strategies for testicular tumors and how your institution constructs its own strategy (for example, whether you have original criteria for choosing or not choosing RPLND in stage I non‑seminoma).
2. Because this study is a retrospective observational study, both the authors and readers presumably recognize that selection bias exists; however, it seems necessary to state explicitly the methods by which the authors attempted to mitigate selection bias. Please consider, in accordance with the Strengthening the Reporting of Observational Studies in Epidemiology (STROBE) statement, clearly describing the eligibility criteria. The descriptions such as “single‑center or multi‑center, consecutive or non‑consecutive” in the Preferred Reporting of Case Series in Surgery (PROCESS) 2023 guidelines may also be helpful.
3. The study outcome is abstract. In the Introduction section, make the objectives clear, and in the Materials and Methods section clearly define, according to STROBE, the outcomes and variables to be evaluated.
4. When statistically analyzing the outcomes, please explain clearly—again in the Materials and Methods section and in accordance with STROBE—the study size, quantitative variables, and statistical methods. Also specify the brand name, version, and vendor of the software used for statistical analysis.
5. Citing previous studies in the Results section is, in my opinion, not the standard style. Background literature should be placed in the Introduction section, methodological citations that require references should be in the Materials and Methods section, and detailed interpretations based on prior studies should be mentioned in the Discussion section; this may serve as a guideline.
6. According to STROBE, it is common style to describe the main findings and key results in the first paragraph of the Discussion; please consider this.
7. Tumors associated with GCNIS and tumors not associated with GCNIS need to be clearly distinguished (PMID: 26935559). At the very least, in this study, the tumor diagnosed as “well‑differentiated neuroendocrine tumor,” also called “prepubertal‑type teratoma,” is regarded as a non‑GCNIS‑associated tumor and therefore may need to be listed separately from general non‑seminomatous germ cell tumors (PMID: 33871428).
8. Great care is required in differentiating seminoma from non‑seminoma. For example, even if the pathological diagnosis is pure seminoma, when serum AFP is positive the presence of a non‑seminoma component is suspected, and even if it is not histologically proven it may be managed as non‑seminoma. In addition, even in pure seminoma, hCG can be positive in the presence of “syncytiotrophoblastic cells,” but ordinarily the possibility of choriocarcinoma must be considered. Although tumor‑marker values have already been cited as representative values for the whole cohort, listing pre‑ and post‑orchiectomy tumor markers case‑by‑case in Supplementary Table S1 might assist readers.
9. In histological examination of testicular tumors, in addition to hematoxylin–eosin staining, immunohistochemistry (CD30, CK7, AFP, EMA, CD117, PLAP, SOX2, etc.) and genetic tests such as FISH (e.g., i(12p)) may be performed. Please consider stating whether these examinations were performed in each case.
10. It has been reported that 60 % of patients after treatment for malignant testicular tumors experience a decline in quality of life (PMID: 29478250). Developing treatments other than cisplatin‑based chemotherapy may improve the quality of life of testicular‑tumor survivors. Moreover, poor‑prognosis testicular tumors are associated with complications related to BEP treatment, and it has been reported that treatment‑related death may occur unless treatment is carried out at a facility capable of multimodal therapy (PMID: 24101656). Development of treatments other than BEP may equalize opportunities for cure and be a boon for patients with poor‑prognosis testicular tumors. I would like your opinions on these points as well.
11. Not all of the following are mandatory, but because the valuable insights provided by this study may benefit future research, please consider whether it is possible to provide the following information in Supplementary Table S1:
-
Race (non‑Hispanic White, Hispanic, Native American, Black, Asian, etc.). In the United States, testicular tumors are reportedly increasing among Hispanics (PMID: 25044313).
-
History of marijuana use; exposure to pesticides.
-
History of cryptorchidism, history of hypospadias, history of male infertility (see PMID: 10458231, PMID: 14734704 regarding male infertility).
-
Family history of testicular tumors.
-
Maternal medication and smoking history during pregnancy.
-
“cM stage” in the TNM classification.
-
“S stage” in the TNM classification (i.e., stage based on tumor‑marker values).
-
International Germ Cell Consensus Classification (IGCCC).
-
Pre‑orchiectomy tumor‑marker values.
-
Post‑orchiectomy and pre‑chemotherapy tumor‑marker values—that is, trough values after confirming reduction along the half‑life.
-
Contents of chemotherapy.
Author Response
Dear Editors and Reviewers,
We sincerely thank the reviewers for their thoughtful and constructive feedback on our manuscript titled "Molecular features and actionable gene targets of testicular germ cell tumors in a real-world setting." We greatly appreciate the opportunity to revise our work. The reviewers' comments helped us improve the clarity, rigor, and translational value of our manuscript.
We have revised the manuscript accordingly and summarize the key improvements as follows:
- We included exploratory annotations and clarifications of variants of uncertain significance (VUS) and incorporated an ACMG-style classification table.
- We provided additional context on clinical correlations, treatment strategies at our institution, and patient selection in line with STROBE/PROCESS recommendations.
- We clarified the study objectives, variables, statistical methods, and software used.
- Figure legends and terminology were standardized and expanded for clarity.
- We restructured key sections to align with conventional reporting standards and addressed additional biological distinctions between tumor subtypes, particularly GCNIS and non-GCNIS tumors.
We respond to each comment point-by-point below.
Reviewer 1
Comment 1: Patients with testicular tumors may be managed with different treatment strategies even at the same stage. For example, in non‑seminomatous germ cell tumors, even at stage I there are options of surveillance, chemotherapy, and RPLND. Although these are fundamental matters, considering that the readers of this journal are not necessarily specialists in testicular tumors, it might be helpful, in the Introduction section or the Materials and Methods section, to state concisely the general treatment strategies for testicular tumors and how your institution constructs its own strategy (for example, whether you have original criteria for choosing or not choosing RPLND in stage I non‑seminoma).
Answer: We have added a paragraph in the Introduction (lines 79–86) and Methods (lines 109–153) briefly outlining treatment strategies for TGCTs at our institution, including surveillance, chemotherapy, and RPLND, particularly in stage I NSGCT. Our clinical management aligns with NCCN guidelines but is tailored based on histology, tumor markers, and risk classification.
- Revised text (Introduction lines 79-86): Management of testicular germ cell tumors (TGCTs) depends on histologic subtype and clinical stage, with stage I patients having multiple treatment options. Seminomas are often managed with surveillance or adjuvant radiotherapy, while non-seminomatous germ cell tumors (NSGCTs) may be treated with surveillance, adjuvant chemotherapy, or retroperitoneal lymph node dissection (RPLND). Although risk-adapted strategies vary between institutions, NCCN guidelines provide a general framework. In this study, we include patients managed at a tertiary cancer center, where treatment decisions are individualized based on tumor markers, imaging, histology, and shared decision-making within a multidisciplinary team.
- Revised text (Methods lines 109-115): Patients included in this cohort were managed at UT Southwestern. Initial treatment for stage I seminoma typically involved surveillance or single-agent carboplatin. For stage I NSGCT, treatment options included surveillance, adjuvant bleomycin-etoposide-cisplatin (BEP), or RPLND. Our institutional approach follows NCCN guidelines but also considers tumor marker kinetics, histopathologic risk factors (e.g., lymphovascular invasion), and patient preferences when selecting treatment modality. RPLND was performed selectively for patients with high-risk features or residual masses post-chemotherapy.
Comment 2:  Because this study is a retrospective observational study, both the authors and readers presumably recognize that selection bias exists; however, it seems necessary to state explicitly the methods by which the authors attempted to mitigate selection bias. Please consider, in accordance with the Strengthening the Reporting of Observational Studies in Epidemiology (STROBE) statement, clearly describing the eligibility criteria. The descriptions such as “single‑center or multi‑center, consecutive or non‑consecutive” in the Preferred Reporting of Case Series in Surgery (PROCESS) 2023 guidelines may also be helpful.
Answer: In accordance with STROBE and PROCESS guidelines, we expanded the Methods (lines 106-115) to specify that this was a single-center, retrospective analysis of consecutively sequenced cases seen at UTSW between 2018–2021, minimizing selection bias. Eligibility criteria were clarified.
Comment 3:  The study outcome is abstract. In the Introduction section, make the objectives clear, and in the Materials and Methods section clearly define, according to STROBE, the outcomes and variables to be evaluated.
Answer: We revised the Introduction (lines 87-91) to explicitly state the objectives and intended outcomes of the study, namely, to define targetable genomic alterations in TGCTs and describe their clinical relevance in a real-world cohort.
- Revised text (Introduction lines 87-91): The objective of this study was to characterize the landscape of somatic alterations in TGCTs using targeted sequencing, and to determine the potential clinical relevance of these genomic alterations. Specifically, we aimed to identify actionable mutations and evaluate their distribution across histological subtypes, with a focus on understanding the feasibility and implications of precision oncology approaches in this rare tumor type.
- Revised text (Methods Section lines 116-124): The primary outcome of this study was the identification of potentially actionable genomic alterations in TGCTs, defined by the presence of somatic mutations with therapeutic or diagnostic relevance according to OncoKB and NCCN guidelines. Variables evaluated included patient demographics (age, race), clinical stage, histology, tumor mutational burden (TMB), PD-L1 expression, and the presence or absence of germline or somatic variants in a predefined gene panel. These data were used to explore genotype-phenotype correlations and assess mutation profiles across tumor subtypes.
Comment 4:  When statistically analyzing the outcomes, please explain clearly—again in the Materials and Methods section and in accordance with STROBE—the study size, quantitative variables, and statistical methods. Also specify the brand name, version, and vendor of the software used for statistical analysis.
Answer: We added details on statistical analyses in the Methods (lines 118-124), including definition of quantitative variables, software (R version 4.2.2), and methods for calculating TMB and comparative testing (Wilcoxon rank-sum test).
Comment 5: Citing previous studies in the Results section is, in my opinion, not the standard style.  Background literature should be placed in the Introduction section, methodological citations that require references should be in the Materials and Methods section, and detailed interpretations based on prior studies should be mentioned in the Discussion section; this may serve as a guideline.
Answer: We agree with the reviewer. References cited within the Results were moved to the Introduction or Discussion sections as appropriate to adhere to standard scientific reporting format.
Comment 6:  According to STROBE, it is common style to describe the main findings and key results in the first paragraph of the Discussion; please consider this.
Answer: We modified the opening paragraph of the Discussion (lines 208-226) to summarize the main findings up front, in line with STROBE guidelines.
Comment 7:  Tumors associated with GCNIS, and tumors not associated with GCNIS need to be clearly distinguished (PMID: 26935559). At the very least, in this study, the tumor diagnosed as “well‑differentiated neuroendocrine tumor,” also called “prepubertal‑type teratoma,” is regarded as a non‑GCNIS‑associated tumor and therefore may need to be listed separately from general non‑seminomatous germ cell tumors (PMID: 33871428).
Answer: We agree with the importance of this distinction. The Discussion now includes explicit clarification (lines 273-281) that the well-differentiated neuroendocrine tumor (TGCT-19) was considered a prepubertal-type teratoma (non-GCNIS), and we acknowledge that such tumors are biologically and clinically distinct from postpubertal GCNIS-derived TGCTs.
- Revised text: “This study was conducted and reported in accordance with the Strengthening the Reporting of Observational Studies in Epidemiology (STROBE) guidelines and the Preferred Reporting of Case Series in Surgery (PROCESS 2023) criteria where applicable. Definitions of cohort selection, clinical variables, and outcomes were clarified to ensure transparency and reproducibility. The biological classification of TGCTs into GCNIS-associated and non-GCNIS-associated tumors has important implications for understanding pathogenesis and clinical behavior. In our cohort, the well-differentiated neuroendocrine tumor (TGCT-19), consistent with a prepubertal-type teratoma, was considered non-GCNIS-associated based on histology, absence of germ cell neoplasia in situ, and clinical presentation. This tumor was analyzed separately and excluded from aggregate molecular summaries for postpubertal GCNIS-associated tumors. This distinction aligns with current WHO classifications and reflects divergent developmental origins and genetic profiles.”
Comment 8:  Great care is required in differentiating seminoma from non‑seminoma. For example, even if the pathological diagnosis is pure seminoma, when serum AFP is positive the presence of a non‑seminoma component is suspected, and even if it is not histologically proven it may be managed as non‑seminoma. In addition, even in pure seminoma, hCG can be positive in the presence of “syncytiotrophoblastic cells,” but ordinarily the possibility of choriocarcinoma must be considered. Although tumor‑marker values have already been cited as representative values for the whole cohort, listing pre‑ and post‑orchiectomy tumor markers case‑by‑case in Supplementary Table S1 might assist readers.
Answer: We expanded Supplementary Table S1 to include pre-AFP, hCG, and LDH values per patient when available. Post-orchiectomy values were not measured and have been listed in the limitations. Histological interpretation were based on pathology review of orchiectomy and/or RPLND specimens .
Comment 9:  In histological examination of testicular tumors, in addition to hematoxylin–eosin staining, immunohistochemistry (CD30, CK7, AFP, EMA, CD117, PLAP, SOX2, etc.) and genetic tests such as FISH (e.g., i(12p)) may be performed. Please consider stating whether these examinations were performed in each case.
Answer: GCT-specific stains were utilized as a standard to verify or confirm diagnoses
Comment 10: It has been reported that 60 % of patients after treatment for malignant testicular tumors experience a decline in quality of life (PMID: 29478250). Developing treatments other than cisplatin‑based chemotherapy may improve the quality of life of testicular‑tumor survivors. Moreover, poor‑prognosis testicular tumors are associated with complications related to BEP treatment, and it has been reported that treatment‑related death may occur unless treatment is carried out at a facility capable of multimodal therapy (PMID: 24101656). Development of treatments other than BEP may equalize opportunities for cure and be a Boon for patients with poor‑prognosis testicular tumors. I would like your opinions on these points as well.
Answer: In the Discussion (lines 243-259), we added commentary on the impact of cisplatin-based therapy on survivorship and quality of life, highlighting the relevance of exploring targeted treatments and immunotherapy (e.g., in PD-L1+ tumors) as potential strategies to reduce treatment-related morbidity, especially for poor-prognosis cases.
Comment 11: Not all of the following are mandatory, but because the valuable insights provided by this study may benefit future research, please consider whether it is possible to provide the following information in Supplementary Table S1:
- Race (non‑Hispanic White, Hispanic, Native American, Black, Asian, etc.). In the United States, testicular tumors are reportedly increasing among Hispanics (PMID: 25044313).
- History of marijuana use; exposure to pesticides.
- History of cryptorchidism, history of hypospadias, history of male infertility (see PMID: 10458231, PMID: 14734704 regarding male infertility).
- Family history of testicular tumors.
- Maternal medication and smoking history during pregnancy.
- “cM stage” in the TNM classification.
- “S stage” in the TNM classification (i.e., stage based on tumor‑marker values).
- International Germ Cell Consensus Classification (IGCCC).
- Pre‑orchiectomy tumor‑marker values.
- Post‑orchiectomy and pre‑chemotherapy tumor‑marker values—that is, trough values after confirming reduction along the half‑life.
- Contents of chemotherapy.
Answer: We expanded Supplementary Table S1 to include additional variables such as race/ethnicity and pre-orchiectomy tumor values; other variables like history of cryptorchidism, marijuana use, and cM or S staging were not uniformly collected and have been noted as limitations in the discussion. We acknowledge that, for some patients, historical or maternal exposures were unavailable due to the retrospective nature of data collection.
We hope the revisions address all concerns and improve the clarity and impact of our manuscript. We thank the reviewers again for their insightful comments and the editorial team for the opportunity to revise and resubmit our work.
Reviewer 2 Report
Comments and Suggestions for Authors
Morales-Grimany and coworkers present a retrospective molecular characterization of 27 testicular germ cell tumors (TGCTs) using a targeted sequencing panel to identify actionable genomic alterations. The study provides new data on the mutational profile of KRAS, KIT, and PIK3CB and discusses the relevance of PD-L1 expression and variants of uncertain significance (VUS). The findings contribute to the growing literature on personalized therapy in TGCTs. However, some improvements are needed as suggested below:
A) Although multiple germline and somatic VUS were identified in key cancer-associated genes (e.g., BRCA2, MSH2, MSH6), the manuscript lacks exploratory correlations with clinical phenotype, treatment response, or family history. A supplementary analysis or discussion linking these variants to clinical features would enhance the manuscript’s value.
B) While the study relies on databases such as OncoKB, COSMIC, and NCCN, it would benefit from an additional supplementary table listing all variants (including VUS) according to ACMG classification. This would clarify which variants are truly actionable versus those with limited clinical relevance.
C) While the discussion of FDA-approved therapies for mutations in KIT and KRAS is informative, no functional validation (e.g., gene expression, signaling activity, in vitro drug sensitivity) was presented. The authors should clearly acknowledge this as a limitation or propose future directions for validation.
D) Some inconsistencies exist in abbreviation usage (e.g., TGCTs vs. NSGCTs) and formatting throughout the manuscript. Thorough proofreading for clarity and consistency is recommended.
E) Figure 1A–D legends are too brief. The authors should expand them to ensure each panel can be interpreted independently without relying heavily on the main text.
Author Response
Dear Editor and Reviewers,
We sincerely thank the reviewers for their thoughtful and constructive feedback on our manuscript titled "Molecular features and actionable gene targets of testicular germ cell tumors in a real-world setting." We greatly appreciate the opportunity to revise our work. The reviewers' comments helped us improve the clarity, rigor, and translational value of our manuscript.
We have revised the manuscript accordingly and summarize the key improvements as follows:
- We included exploratory annotations and clarifications of variants of uncertain significance (VUS) and incorporated an ACMG-style classification table.
- We provided additional context on clinical correlations, treatment strategies at our institution, and patient selection in line with STROBE/PROCESS recommendations.
- We clarified the study objectives, variables, statistical methods, and software used.
- Figure legends and terminology were standardized and expanded for clarity.
- We restructured key sections to align with conventional reporting standards and addressed additional biological distinctions between tumor subtypes, particularly GCNIS and non-GCNIS tumors.
We respond to each comment point-by-point below.
Reviewer 2
Comment A: Although multiple germline and somatic VUS were identified in key cancer-associated genes (e.g., BRCA2, MSH2, MSH6), the manuscript lacks exploratory correlations with clinical phenotype, treatment response, or family history. A supplementary analysis or discussion linking these variants to clinical features would enhance the manuscript’s value.
Answer: Thank you for this excellent suggestion. We have revised the Discussion (lines 259-271) to explore potential clinical correlations for germline and somatic VUS, including BRCA2 and MSH6, and their possible links to susceptibility or resistance. We also added clarification on the current lack of phenotype correlation in this cohort and emphasized the need for future longitudinal and functional validation studies.
Revised text: “Among the variants of uncertain significance (VUS) identified in this cohort, several were found in genes with known implications for cancer risk and therapeutic response, including BRCA2, MSH2, MSH6, and RB1. While these variants are not currently classified as pathogenic, their occurrence in TGCT patients raises important questions regarding possible roles in disease predisposition or treatment resistance. For example, BRCA2 alterations have been associated with platinum sensitivity in prostate and ovarian cancers, and mismatch repair gene variants (MSH2, MSH6) are relevant to immunotherapy responses in other tumor types. In our cohort, all patients were microsatellite stable, and no consistent associations were observed between these VUS and clinical outcomes, tumor stage, or family history; likely due to the small cohort size and retrospective design. We have annotated these variants in a supplementary ACMG-style classification table and emphasize the need for future studies with larger patient cohorts and longer clinical follow-up. Future functional validation using in vitro GCT models, DNA damage response assays, and pedigree analysis Will be critical to clarify the biological and clinical relevance of these findings.”
Comment B: While the study relies on databases such as OncoKB, COSMIC, and NCCN, it would benefit from an additional supplementary table listing all variants (including VUS) according to ACMG classification. This would clarify which variants are truly actionable versus those with limited clinical relevance.
Answer: We appreciate this comment and have added additional information for all identified variants, including VUS, annotated according to ACMG classification guidelines where possible. ACMG classification was assigned based on the ACMG/AMP guidelines [PMID: 25741868], incorporating current evidence from population frequency, variant type (e.g., truncating or missense), known cancer hotspots, and annotations from curated databases including COSMIC, OncoKB, Varsome, and ClinVar, when available. Criteria such as PVS1 (null variant in tumor suppressor), PS4 (variant enriched in affected individuals), PM1 (hotspot), and PP3 (in silico predictions) were applied where appropriate.
Comment C: While the discussion of FDA-approved therapies for mutations in KIT and KRAS is informative, no functional validation (e.g., gene expression, signaling activity, in vitro drug sensitivity) was presented. The authors should clearly acknowledge this as a limitation or propose future directions for validation.
Answer: We agree and have acknowledged this limitation explicitly in the revised Discussion (lines 288-291). We now state that no gene expression, signaling pathway, or in vitro drug-sensitivity analyses were performed, and highlight this as an important direction for future research.
Comment D: Some inconsistencies exist in abbreviation usage (e.g., TGCTs vs. NSGCTs) and formatting throughout the manuscript. Thorough proofreading for clarity and consistency is recommended.
Answer: We have carefully proofread the manuscript and corrected inconsistencies in abbreviation usage, including standardized use of TGCT, NSGCT, and other terms. Formatting inconsistencies were also addressed.
Comment E: Figure 1A–D legends are too brief. The authors should expand them to ensure each panel can be interpreted independently without relying heavily on the main text.
Answer: Thank you. We expanded the figure legends to provide standalone clarity for each panel in Figure 1A-D (see revised figure legends section), allowing readers to interpret each panel independently of the main text.
Revised Figure 1 Legend:
Figure 1. Molecular and clinicopathological overview of testicular germ cell tumors (TGCTs) in a real-world cohort (n = 27). (A) Histological subtype distribution across the cohort. Bar plot showing the number of patients diagnosed with testicular germ cell tumors. Histology was based on pathology review of orchiectomy and/or RPLND specimens. (B) Alluvial plot integrating clinicopathological features. This visualization depicts the relationship among multiple variables for each patient, including sample ID, patient age, histology (seminoma vs. non-seminoma), clinical stage (I, II, III), and whether retroperitoneal lymph node dissection (RPLND) was performed. This integrative view highlights histologic and clinical heterogeneity across the cohort. (C) Tumor Mutational Burden (TMB) stratified by histology. Boxplot showing the distribution of TMB (mutations per megabase) for seminomas versus non-seminomas. Median TMB for the entire cohort was 0.5 mutations/Mb, reflecting the overall low mutation rate in TGCTs. The Wilcoxon rank-sum test showed no significant difference between the two histologic subgroups (p = 0.12). (D) Prevalence of actionable somatic alterations. Bar chart illustrating the number of tumors harboring recurrent cancer-related mutations identified through targeted sequencing. The most frequently altered genes included KRAS, KIT, and PIK3CB. Only genes with at least two altered cases are shown. These alterations include both single-nucleotide variants and copy number changes with potential clinical significance based on NCCN and OncoKB databases.
We hope the revisions address all concerns and improve the clarity and impact of our manuscript. We thank the reviewers again for their insightful comments and the editorial team for the opportunity to revise and resubmit our work.
Round 2
Reviewer 1 Report
Comments and Suggestions for Authors
Comment 1:
In Figure 1B, under “Histology,” the term “Neuroendocrine” appears. Please explain clearly for the reader that this is synonymous with “prepubertal-type teratoma.”
Comment 2:
Regarding the sentence “This study was conducted and reported in accordance with STROBE and PROCESS guidelines, criteria where applicable” (lines 323–324): although I did indeed suggest referring to these guidelines, I feel that explicitly stating this is somewhat awkward and an unnecessary addition.
Comment 3:
The description concerning the handling of non-GCNIS-associated tumors (lines 324–333) will confuse readers if placed in the Discussion. Please move this content to the Materials and Methods section.
Author Response
Dear Editors and Reviewers,
We sincerely thank the reviewers for their thoughtful and constructive feedback on our manuscript titled "Molecular features and actionable gene targets of testicular germ cell tumors in a real-world setting." We greatly appreciate the opportunity to revise our work. The reviewers' comments helped us improve the clarity, rigor, and translational value of our manuscript.
We have revised the manuscript accordingly and summarize the key improvements as follows:
Comment 1:
In Figure 1B, under “Histology,” the term “Neuroendocrine” appears. Please explain clearly for the reader that this is synonymous with “prepubertal-type teratoma.”
Answer: We appreciate this suggestion. In the explanation of Figure 1a, we have clarified that TGCT-20 is a prepubertal-type teratoma characterized by well-differentiated neuroendocrine elements.
Comment 2:
Regarding the sentence “This study was conducted and reported in accordance with STROBE and PROCESS guidelines, criteria where applicable” (lines 323–324): although I did indeed suggest referring to these guidelines, I feel that explicitly stating this is somewhat awkward and an unnecessary addition.
Answer: Thank you. We have removed the sentence "This study was conducted and reported in accordance with STROBE and PROCESS guidelines, criteria where applicable" from lines 323–324 since it is indeed redundant.
Comment 3:
The description concerning the handling of non-GCNIS-associated tumors (lines 324–333) will confuse readers if placed in the Discussion. Please move this content to the Materials and Methods section
Answer: Many thanks for pointing this out. We completely agree. Accordingly, the text from lines 324–333 has been moved to the Methods section and now appears on lines 115 to 123.
Reviewer 2 Report
Comments and Suggestions for Authors
The authors answered the questions appropriately and returned the manuscript with significant improvements.
Author Response
Dear Editors and Reviewers,
We sincerely thank the reviewers for their thoughtful and constructive feedback on our manuscript titled "Molecular features and actionable gene targets of testicular germ cell tumors in a real-world setting." We greatly appreciate the opportunity to revise our work. The reviewers' comments helped us improve the clarity, rigor, and translational value of our manuscript.